# Reversal of pathological motor behavior in a model of Parkinson's disease by striatal dopamine uncaging

**Miguel A. Zamora-Ursulo**[1], **Job Perez-Becerra**[2], **Luis A. Tellez**[2], **Nadia Saderi**[1], **Luis Carrillo-Reid**[2]*

**1** Facultad de Ciencias, Universidad Autónoma de San Luis Potosi, San Luis Potosi, Mexico, **2** Instituto de Neurobiologia, Universidad Nacional Autónoma de Mexico, Juriquilla, Queretaro, Mexico

* carrillo.reid@comunidad.unam.mx

**Data Availability Statement:** All relevant data are within the paper and its Supporting information files.

## Abstract

Motor deficits observed in Parkinson's disease (PD) are caused by the loss of dopaminergic neurons and the subsequent dopamine depletion in different brain areas. The most common therapy to treat motor symptoms for patients with this disorder is the systemic intake of L-DOPA that increases dopamine levels in all the brain, making it difficult to discern the main locus of dopaminergic action in the alleviation of motor control. Caged compounds are molecules with the ability to release neuromodulators locally in temporary controlled conditions using light. In the present study, we measured the turning behavior of unilateral dopamine-depleted mice before and after dopamine uncaging. The optical delivery of dopamine in the striatum of lesioned mice produced contralateral turning behavior that resembled, to a lesser extent, the contralateral turning behavior evoked by a systemic injection of apomorphine. Contralateral turning behavior induced by dopamine uncaging was temporarily tied to the transient elevation of dopamine concentration and was reversed when dopamine decreased to pathological levels. Remarkably, contralateral turning behavior was tuned by changing the power and frequency of light stimulation, opening the possibility to modulate dopamine fluctuations using different light stimulation protocols. Moreover, striatal dopamine uncaging recapitulated the motor effects of a low concentration of systemic L-DOPA, but with better temporal control of dopamine levels. Finally, dopamine uncaging reduced the pathological synchronization of striatal neuronal ensembles that characterize unilateral dopamine-depleted mice. We conclude that optical delivery of dopamine in the striatum resembles the motor effects induced by systemic injection of dopaminergic agonists in unilateral dopamine-depleted mice. Future experiments using this approach could help to elucidate the role of dopamine in different brain nuclei in normal and pathological conditions.

## Introduction

Parkinson's disease (PD) is a devastating neurodegenerative disorder caused by the progressive loss of dopamine in the brain [1]. PD is characterized by motor abnormalities such as bradykinesia, tremor, and posture unbalance [2–4]. Motor dysfunctions in PD emerge after the severe

**Funding:** This research was supported by grants from CONACYT (CF6653, CF154039) and UNAM-DGAPA-PAPIIT (IA201421, IA201819, IN213923) to L.C-R. MA.Z-U. participated in this work in partial fulfillment of the requirements for the Ph.D. degree in Basic Biomedical Sciences at the Universidad Autonoma de San Luis Potosi graduate scholarship from CONACYT (770504). The funders had no role in study design, data collection and analysis, decision to publish, or preparation of the manuscript.

**Competing interests:** NO authors have competing interests.

destruction of dopaminergic neurons of the mesencephalon and the degeneration of their axonal projections to the striatum [5, 6]. The striatum is the main entry gateway of the basal ganglia, and it has been shown that the loss of dopamine evokes abnormal synchronization of striatal neuronal populations [7–11].

Despite that PD doesn't have cure, different treatments are used to alleviate its motor deficits [12]. Among them, L-DOPA remains as the most effective therapy that has been used for over 60 years [12–14]. L-DOPA is a dopaminergic precursor that crosses the blood-brain barrier and chronically increases dopamine levels improving motor symptoms [15, 16]. However, prolonged L-DOPA intake generates dyskinesias [17] that require the adjunct use of dopaminergic agonists causing undesired side effects such as hallucinations or compulsive behaviors [17–19]. The limitations of current therapies for PD highlight the need of pharmacological tools that resemble dopamine fluctuations in physiological conditions [20].

Photopharmacology intends to avoid the side effects caused by pharmacotherapy using probes that are formed by a photosensitive cage attached to the structure of a molecule that is biologically inactive before illumination [21, 22]. Light irradiation detaches the bioactive molecule from the cage allowing its interaction with cellular receptors at high spatiotemporal scales. Thus, caged compounds allow the dynamic control of drug activity avoiding non-desired side effects caused by poor spatial and temporal drug release. Several caged compounds have been used for basic research in animal models [22, 23]; however, their clinical applications for neurological disorders are still to be proven.

The first caged compounds were made using organic chemical reactions that attached a photocleavable group (cage) to a biomolecule [21]. A more recent strategy was the development of ruthenium-based caged compounds that are formed by a metal center of ruthenium-polypyridine with high affinity for amine groups. Ruthenium-Based caged compounds have the ability to deliver biologically active molecules like: 4-aminopyridine, glutamate, gamma aminobutyric acid, glycine, serotonin, dopamine, or nicotine with fast temporal and spatial resolution using visible or infrared light [24]. It has been shown in vitro that a caged dopamine compound (RuBi-Dopa) could be released with high temporal and spatial resolution modulating dopamine receptors in dendritic spines [25]. Furthermore, it has been shown in vivo that dopamine uncaging with visible light modulates the local field potential (LFP) in medial prefrontal cortex of healthy rats [26]. However, the effect of dopamine uncaging in animal models of PD remains unknown.

One of the most used animal models to measure the motor effects caused by the destruction of dopaminergic neurons consists of the unilateral injection of 6-hydroxydopamine (6-OHDA) in the substantia nigra pars compacta (SNc). In rodents, it has been shown that the destruction of dopaminergic neurons in one brain hemisphere causes a movement imbalance reflected as ipsilateral turning behavior toward the dopamine-depleted side of the brain [7, 9–11]. In such model of PD the systemic injection of a dopaminergic agonist (apomorphine) induces contralateral turning behavior [27] suggesting that dopamine uncaging could also induce contralateral turning behavior.

To investigate the effect of dopamine elevation in the lesioned side of unilateral dopamine-depleted mice we uncaged dopamine in the striatum and measured: contralateral turning behavior, dopamine concentration, and striatal population activity.

## Materials and methods

### Animals

Experiments were performed on C57BL/6J male mice, 60–70 postnatal days before surgical procedures. We used 75 mice for experiments and data analyses and discarded 12 animals due

to failures in stereotaxic coordinates to reach the SNc. Mice were housed on a 12 h light-dark cycle with food and water ad libitum. All experimental procedures were carried out in accordance with the guidelines of the Bioethics Committee of the Neurobiology Institute for the care and use of laboratory animals that comply with the standards outlined by the Guide for the Care and Use of Laboratory Animals (NIH).

## Stereotaxic surgeries

Mice were anesthetized with isoflurane (1–2%) and placed in a stereotaxic system (Stoelting Co., IL.). All procedures were performed in sterile conditions. Respiratory rate and tail pinch reflex were monitored along the surgery. For unilateral dopamine-depleted mice, 1μL (5 mg/mL in 0.9% NaCl and 0.5% ascorbate) of the neurotoxin 6-hydroxydopamine (6-OHDA) was slowly injected (0.05 μL/min in the right Substantia nigra pars compacta (SNc) at stereotaxic coordinates (bregma: AP, -3 mm; L, -1.3 mm; and -4.3 mm below dura). Experiments were performed ~3 weeks after 6-OHDA injection. For control non-dopamine-depleted mice, 1μL of saline solution was injected at the same coordinates. On a group of mice, after the intracerebral injection, a 0.5 mm craniotomy was performed on top of the right striatum (AP: 0.7 mm; ML: -1.7 mm; DV: -2.35 mm) to stereotaxically insert a cannula (24 gauge; 9 mm long) that was used to locally inject RuBi-Dopa into the striatum and subsequently introduce a fiber optic cannula for light uncaging (400 μm diameter, 0.39 NA, Thorlabs). On a different group of mice, the cannula described above, and a fiber optic cannula were implanted, with an angle of 34˚ between them, such as that both tips of the cannulas converged. In such experiments the cannula was used to locally inject RuBi-Dopa into the striatum. On a different group or mice, a cannula was implanted to insert the microdialysis probe, to measure dopamine levels in control and unilateral dopamine-depleted mice. On a different group of mice, a microdialysis cannula and a cannula for the fiber optic and RuBi-Dopa injection were implanted, with an angle of 34˚ between them, such as that both tips of the cannulas converged. The cannulas used have a removable dummy protective cap to avoid clogging. Finally, for all the mice a custom designed stainless steel head plate was attached to the skull using dental cement. During surgeries eyes were moisturized with eye ointment. For 5 days after surgery mice received subcutaneously 0.5 ml of saline/glucose (4%) solution to prevent dehydration. The first 5 days after 6-OHDA injection mice were manually fed with chow and liquid supplements to avoid weight loss and promote recovery.

## Tyrosine hydroxylase immunofluorescence

To remove the brain, mice were deeply anesthetized with sodium pentobarbital injected intraperitoneally and intracardially perfused with 4% paraformaldehyde (PFA). Brains were fixed in 4% buffered PFA for 2 days and cryoprotected in 30% sucrose solution. 40 μm thick coronal sections were cut in a cryostat (CM3050S Leica). Brain slices were washed in phosphate-buffered saline (PBS), permeabilized with Citrate/Triton X-buffer (1% sodium citrate; 1% triton X-100) for 15 minutes at room temperature. Then sections were blocked with 5% normal goat serum for 30 minutes and incubated for 72 hours (4˚C) with rabbit polyclonal anti-Tyrosine hydroxylase (TH) antibody (1:1500, ab6211, Abcam). Afterward, brain sections were incubated with goat anti-rabbit IgG H&L (Alexa Fluor 488; 1:1000, ab150077, Abcam) and mounted with DAPI medium (VECTASHIELD PLUS antifade mounting medium with DAPI, H-2000, Vector Laboratories). TH expression was visualized with a confocal microscope Zeiss LSM 780 and Zen software. Mosaic images were stitched to represent a full coronal section of the brain. ImageJ (NIH) was used to quantify fluorescence levels corresponding to TH expression. Fluorescence levels of the dopamine-depleted side of the brain were normalized to the fluorescence levels of the intact side of the brain.

## Pharmacology

All experiments were done after ~21 days of 6-OHDA injection in the SNc. We performed systemic intraperitoneal injections of apomorphine (0.5 mg/kg; Sigma-Aldrich), or L-DOPA (1 mg/kg; or 6 mg/kg; Sigma-Aldrich). L-DOPA was mixed with benserazide (15 mg/kg; Sigma-Aldrich) in 0.9% saline solution. RuBi-Dopa (1.5 μL; 300 μM; 0.3 μL/min; Abcam) was injected locally into the striatum through an implanted cannula.

## Open field arena

To characterize turning behavior in unilateral-dopamine depleted mice, animals were placed in a transparent acrylic square box (42 x 42 x 30 cm) elevated 1.5 m from the ground level. A video camera (PlayStation Eye, Sony) was placed under the acrylic box. We performed 10 min recordings at a frame rate of 60 frames/second for different experimental conditions. Turning behavior was measured 10 minutes after systemic injection of drugs, 10 minutes after dopamine uncaging, or at the same time of dopamine uncaging. To quantify the number of turns in the open field arena DeepLabCut (v.2.2.0; ResNet-50; 500000 training iterations) was used [28]. The nose, body and base of the tail were used as reference points for skeletal representation of the mouse position. Turning behavior was determined by computing the angle between such markers and the distance traveled was computed using the nose as a reference point [29].

## Dopamine uncaging

For RuBi-Dopa injections mice were placed on a custom designed jetball system. An injector needle connected to an infusion pump (Fusion 200, Chemix) was inserted through the cannula in the striatum of the lesioned side of the brain. After RuBi-Dopa injection the needle was removed, and diffusion was allowed for 10 minutes. In the group of mice without a fiber optic cannula implant, a fiber optic cannula attached to a compatible fiber optic and connected to a blue LED (470 nm) was inserted through the injection cannula and RuBi-Dopa was irradiated with light for 5 min using a LED controller (CD2100, Thorlabs, duty cycle 20%, 20Hz, 4 mW). In such group of mice, the fiber optic cannula was removed after dopamine uncaging, and the animals were placed on the open field arena after 10 minutes of dopamine uncaging. In the group of mice with a fiber optic cannula implant, mice were placed on the open field arena after 10 minutes of RuBi-Dopa injection, with the fiber optic cannula attached to a fiber optic connected to a rotary joint (Thorlabs) to allow the movement of the animals with the fiber attached, so that dopamine uncaging cold be performed when the animals were into the open field arena.

## Microdialysis and high-performance liquid chromatography (HPLC)

A microdialysis probe (1mm CMA-7, 6 kDa, CMA) connected to an infusion pump was inserted through the cannula placed in the striatum. Artificial cerebrospinal fluid (ACSF) was perfused at 1.2 μL/min. After the probe insertion we waited for 40 minutes to avoid artifacts evoked by mechanical manipulation. Each sample was collected for 5 minutes in awake animals moving freely on a custom designed jetball system. Immediately after collection, each sample was quantified by an HPLC system (Eicom). Chromatograms were analyzed with the software EPC-300 (Eicom). Dopamine concentration was determined using a dopamine solution of 0.5 pg/μL (Sigma-Aldrich). The temporal course of uncaged dopamine was normalized to the maximum peak evoked by LED irradiation. We used such normalization because the measurements of uncaged dopamine in the samples varied as a function of the distance between the fiber optic and the microdialysis probe.

## Electrophysiology

To perform local field potential (LFP) recordings of striatal populations a 3mm diameter craniotomy was done over the striatum of anesthetized dopamine-depleted mice (urethane 1 g/kg). An injector attached to an optic fiber cannula connected to a fiber optic and a LED were inserted at 30˚ on the dopamine-depleted side of the brain. A Silicon probe (Neuronexus, A4x4-tet-5mm) was inserted vertically (AP: 0.5 mm; ML: -2.5 mm; DV: -3.2 mm) until it converged with the fiber optic cannula. LFPs were acquired with OmniPlex Neural recording data acquisition system (Plexon) and low pass filtered (<300 Hz). For bursting analysis, the continuous wavelet transform (CWT) was applied to filtered LFP recordings. CWTs are used to factorize signals with sudden transitions that are not well described by Fourier analysis [30]. For time-frequency analysis a Morlet wavelet was used (1–200 Hz). Bursts were defined as events with amplitude >1 S.D of the CWT. The amplitude of each burst was normalized to the maximum peak of each recording. The burst duration represents the interval of each burst at half amplitude. The burst interval was measured between the peaks of each adjacent burst.

## Analyses and statistical methods

We did not use statistical power analysis to determine the number of animals used in each experiment. We determined the sample size following previous publications [11]. All values in the text stated mean ± S.D. Male mice littermates were randomly assigned to experimental groups before surgeries. Experimental data were collected not blinded to experimental groups. MATLAB R2021b (MathWorks) was used for data analysis. Statistical tests were done in Graphpad Prism. Statistical details of each experimental group can be found in figure legends. One-tail tests were performed in all experiments. Data presented as whisker boxplots display: median, interquartile and range values.

# Results

## Characterization of turning behavior in unilateral dopamine-depleted mice

To characterize the turning behavior described in unilateral dopamine-depleted mice, we injected 6-OHDA into the right SNc on one group of mice (dopamine-depleted), or saline solution in another group of mice (control) (Fig 1A). After 3 weeks of 6-OHDA injection in the SNc, we observed a decrease of the dopaminergic innervation to the striatum of the lesioned side of the brain (Fig 1B) corresponding to ~80% loss of the dopaminergic terminals (Fig 1C; normalized TH fluorescence of lesioned side: 19.74% ± 4.135%). Control animals (saline injected unilaterally in the SNc) placed inside an open field arena moved equally around all the borders of the box (Fig 1D), whereas 6-OHDA injected mice showed restrained mobility inside the box (Fig 1E), that was reflected as a reduction of the distance traveled (distance traveled control: 5590 ± 886 cm; distance traveled lesioned: 4319 ± 1137; *P = 0.0325; Mann Whitney test; n = 6 mice). Control animals moved without a preference to display turning behavior toward the right or the left direction (Fig 1F; turns to the right: 0.0833 ± 0.0753; turns to the left: 0.0667 ± 0.1033), whereas dopamine depleted mice displayed ipsilateral turning behavior (S1 Video) toward the lesioned side of the brain (Fig 1F; ipsilateral turns to the lesioned side: 4.35 ± 0.6775; contralateral turns to the lesioned side: 0 ± 0). Our experiments confirm that unilateral dopamine-depleted mice could allow the characterization of turning behavior under different pharmacological conditions.

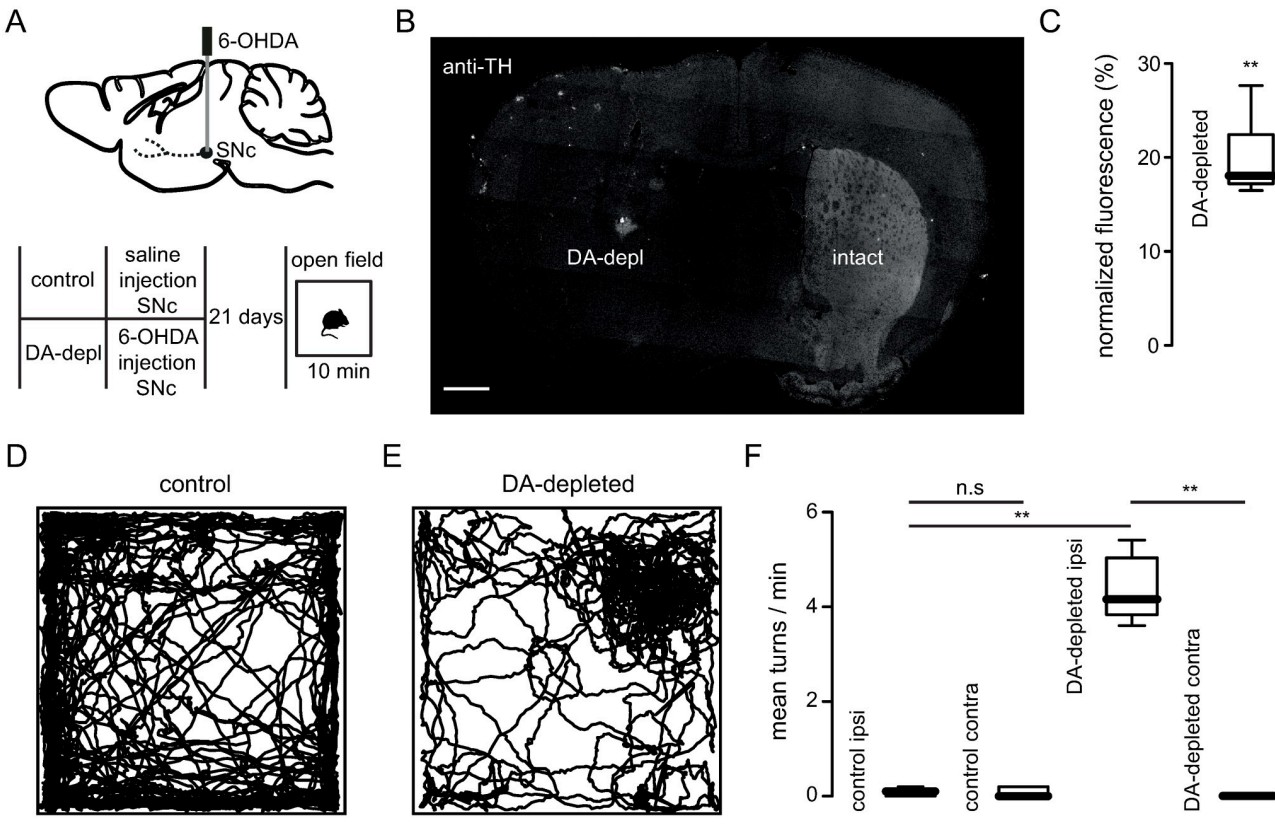

**Fig 1. Ipsilateral turning behavior in unilateral dopamine-depleted mice.** (A) Schematic representation of a sagittal brain section showing the 6-OHDA injection on SNc (top), and the experimental time line (bottom). (B) Tyrosine hydroxylase (TH) immunofluorescence of a representative coronal brain slice showing the dopaminergic terminals in the striatum of intact and dopamine-depleted hemispheres. Scale bar: 500 μm. The gray rectangle on the lesioned side depicts the fiber optic trajectory. Note the lack of TH staining (fluorescence) on the dopamine-depleted side of the brain. (C) Percentage of normalized fluorescence with respect to the intact side showing the fluorescence levels in the dopamine-depleted side (**p = 0.0014; n = 6 mice; Mann-Whitney test). (D) Movement trajectory of a representative control mouse (saline injected in SNc) placed on an open field arena. Note that mice moved similarly at different places inside the box. Scale bar: 10cm. (E) Movement trajectory of a representative unilateral dopamine-depleted mouse (6-OHDA injected in SNc) placed on an open field arena. Note that lesioned mice have restricted movement due to turning behavior. Scale bar: 10cm. (F) Non-lesioned mice didn't show a preference for ipsilateral (ipsi) or contralateral (contra) turns to the lesioned side (p = 0.3320; n = 6 mice; Mann-Whitney test). Unilateral dopamine-depleted mice showed ipsilateral turning behavior to the lesioned side (**p = 0.0014; n = 6 mice; Mann-Whitney test). The number of ipsilateral turns is significantly different between control and unilateral dopamine-depleted mice (**p = 0.0023; n = 6 mice; Mann-Whitney test).

## Contralateral turning behavior induced after striatal dopamine uncaging in unilateral dopamine-depleted mice

The unilateral dopamine-depleted experimental model of PD has been broadly used for pharmacological studies aiming to characterize the effects of different neuromodulators on motor behavior [9, 11, 31]. In such model of PD, it has been shown that the systemic injection of dopaminergic agonists induced contralateral turning behavior toward the lesioned side of the brain [27]. To characterize the contralateral turning behavior in unilateral dopamine-depleted mice we injected apomorphine systemically and observed that after 10 minutes of the injection (Fig 2A and S2 Video) lesioned mice placed inside an open field arena switched from ipsilateral to contralateral turning behavior (ipsilateral turns to the lesioned side before apomorphine: 4.183 ± 0.4535; contralateral turns to the lesioned side before apomorphine: 0 ± 0) that restrained their movement inside the box (Fig 2B) reflected as a reduction of the distance traveled (distance traveled before apomorphine: 3791 ± 490 cm; distance traveled after

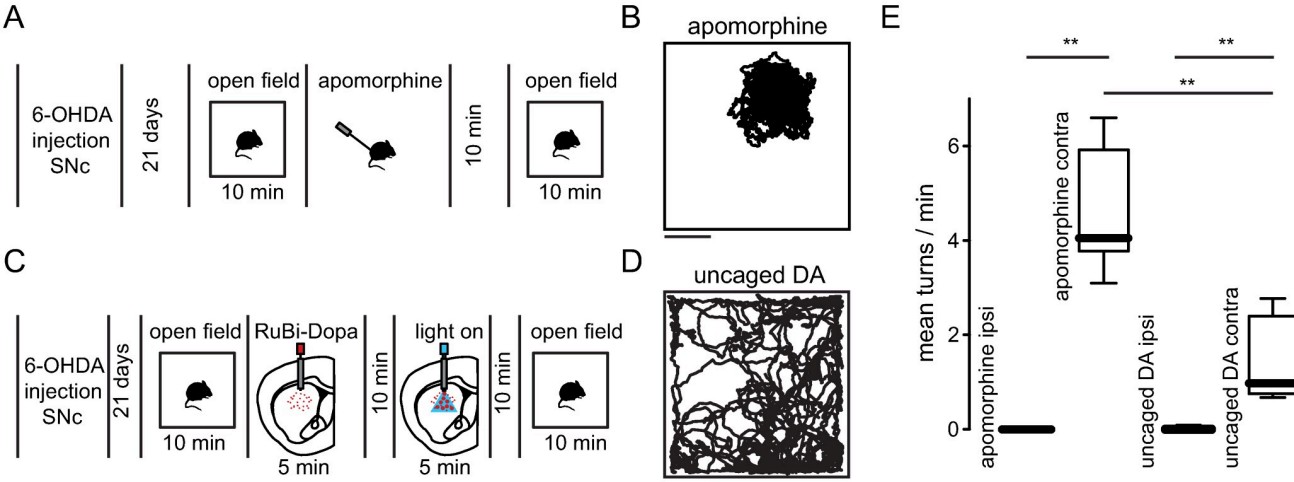

**Fig 2. Dopamine uncaging induced contralateral turning behavior in unilateral dopamine-depleted mice.** (A) Experimental timeline of systemically injected apomorphine in unilateral dopamine-depleted mice. (B) Movement trajectory of a representative unilateral dopamine-depleted mouse after 10 minutes of systemic injection of apomorphine. Note the restriction of movement due to turning behavior. Scale bar: 10cm. (C) Experimental timeline of striatal injection of RuBi-Dopa in unilateral dopamine-depleted mice. (D) Movement trajectory of a representative unilateral dopamine-depleted mouse after 10 minutes of striatal dopamine uncaging. Note the restriction of movement due to turning behavior. Scale bar: 10cm. (E) Unilateral dopamine-depleted mice systemically injected with apomorphine showed contralateral turning behavior to the lesioned side (**p = 0.0014; n = 6 mice; Mann-Whitney test). Lesioned mice showed contralateral turning behavior to the lesioned side evoked by dopamine uncaging in the striatum (**p = 0.0018; n = 6 mice; Mann-Whitney test). Striatal dopamine uncaging produced fewer contralateral turns than systemic apomorphine injection (**p = 0.0025; n = 6 mice; Mann-Whitney test).

apomorphine: 2783 ± 457; *P = 0.0313; Wilcoxon matched-pairs signed rank test; n = 6 mice). Control animals (saline injected unilaterally in the SNc) didn't display turning behavior after apomorphine injection (S1 Fig). On a different group of mice, we injected RuBi-Dopa locally into the striatum of the lesioned side of the brain and then inserted a fiber optic cannula attached to a blue LED to optically release dopamine into the striatum for 5 minutes (470 nm, 20Hz, 4 mW, 20% duty cycle). Afterwards, to compare the motor effects of dopamine uncaging with the systemic injection of apomorphine we waited 10 minutes after the light stimulation protocol and measured the motor behavior (Fig 2C and S3 Video). We observed that dopamine uncaging also produced a switch from ipsilateral to contralateral turning behavior (ipsilateral turns to the lesioned side before dopamine uncaging: 3.683 ± 0.7985; contralateral turns to the lesioned side before dopamine uncaging: 0.0167 ± 0.0408) that restricted the movement of mice in an open field arena, but in a lower degree (Fig 2D). Accordingly, the distance traveled after dopamine uncaging was further reduced (distance traveled before dopamine uncaging: 4363 ± 1281 cm; distance traveled after dopamine uncaging: 1481 ± 773; *P = 0.0156; Wilcoxon matched-pairs signed rank test; n = 6 mice). Unilateral dopamine-depleted mice systemically injected with apomorphine displayed contralateral turning behavior instead of ipsilateral turning behavior (Fig 2E; ipsilateral turns to the lesioned side: 0 ± 0; contralateral turns to the lesioned side: 4.583 ± 1.298). Comparably, striatal dopamine uncaging produced, to a lesser extent, more contralateral turns than ipsilateral turns (Fig 2E; ipsilateral turns to the lesioned side: 0.0167 ± 0.0408; contralateral turns to the lesioned side: 1.433 ± 0.8869). Furthermore, striatal dopamine uncaging induced significantly fewer contralateral turns than the systemic injection of apomorphine (Fig 2E; contralateral turns between apomorphine vs. dopamine uncaging: **p = 0.0049; Mann Whitney test; n = 6 mice). Control animals (saline injected unilaterally in the SNc) didn't display turning behavior after dopamine uncaging (S1 Fig; p = 0.2023; n = 6 mice; Mann-Whitney test; ipsilateral turns to the lesioned side: 0 ± 0;

contralateral turns to the lesioned side: 0.0333 ± 0.0816). Moreover, we observed that the contralateral turning behavior in lesioned mice, induced by optical release of dopamine returned to ipsilateral turning behavior after one hour, without further dopamine release, suggesting that the levels of dopamine were increased by uncaging and then returned to basal conditions.

### Temporal course of dopamine levels and contralateral turning behavior after dopamine uncaging

To measure dopamine levels in the striatum we performed microdialysis and High-performance liquid chromatography (HPLC) in control and unilateral dopamine-depleted mice (Fig 3A). Compared to non-lesioned mice, dopamine-depleted mice showed decreased levels of dopamine corroborating the destruction of dopaminergic neurons (Fig 3B; dopamine concentration control mice: 0.3738 ± 0.0726 pg/μL; dopamine concentration unilateral dopamine-depleted mice lesioned side: 0.0983 ± .0303 pg/μL). Since we previously observed that contralateral turning behavior induced by dopamine uncaging lasted around one hour, we measured the dopamine levels in lesioned mice at different times after the optical release of dopamine (Fig 3C) and observed that after 60 minutes the peak of dopamine evoked by light stimulation returned to basal conditions (Fig 3D; normalized dopamine at different times, basal: 1.327 ± 2.058%; 10 min: 48.28 ± 13.79%; 30 min: 15.66 ± 20.73%; 60 min: 2.736 ± 4.812%)

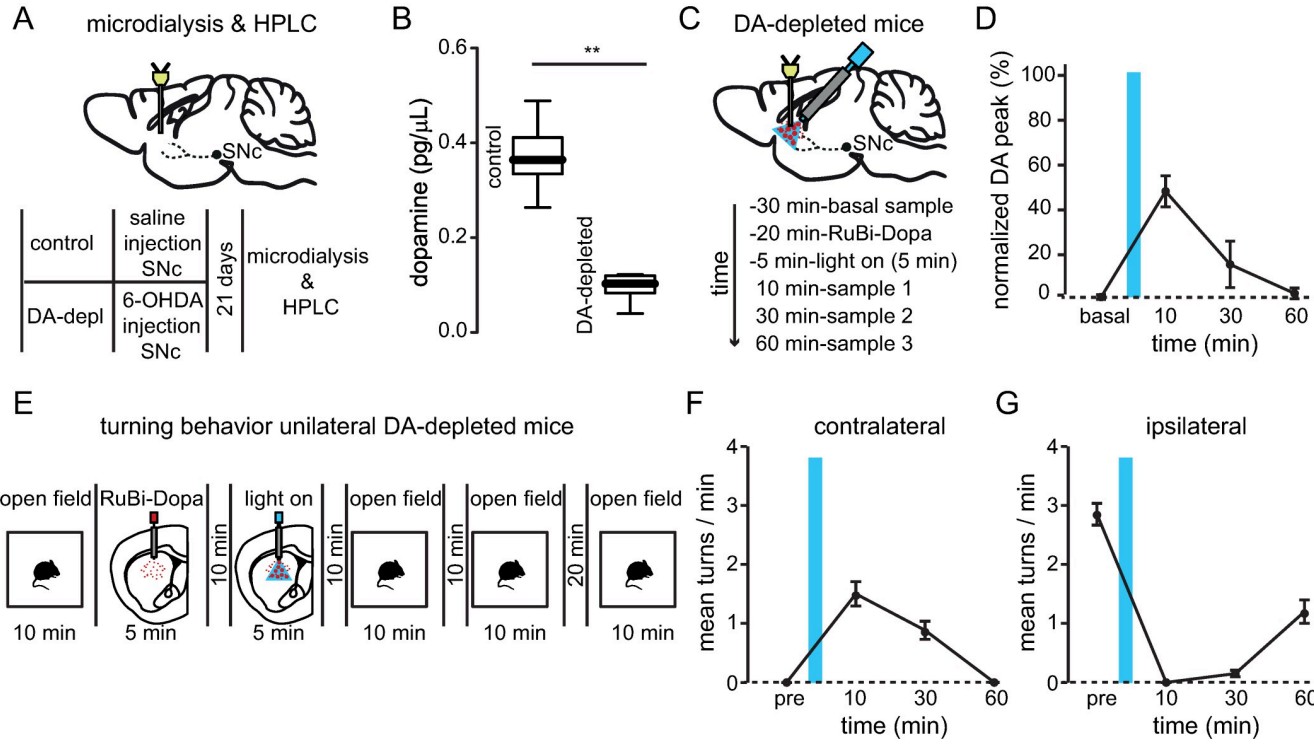

**Fig 3. Dopamine levels and turning behavior in unilateral dopamine-depleted mice.** (A) Schematic representation of microdialysis experiments in control and unilateral dopamine-depleted mice. (B) Dopamine measured by microdialysis and HPLC showed a reduced concentration of unilateral dopamine-depleted animals compared to control animals (**p = 0.0011; n = 6 mice; Mann-Whitney test). (C) Schematic representation of microdialysis experiments to measure uncaged dopamine in lesioned mice. (D) Normalized dopamine levels evoked by dopamine uncaging in unilateral dopamine-depleted mice before and after light stimulation. Time courses display mean ± s.e.m. (n = 4 mice). Note that dopamine levels return to basal conditions after 60 minutes. (E) Schematic representation of the temporal course of turning behavior evoked by dopamine uncaging in unilateral dopamine-depleted mice. (F) Contralateral turning behavior evoked by dopamine uncaging followed a similar temporal course of dopamine levels. Note that after 60 minutes the contralateral turning behavior returned to pre-stimulation conditions. (G) Ipsilateral turning behavior depended on dopamine levels. Note that after 60 minutes the ipsilateral turning behavior is reinstated. Time courses display mean ± s.e.m. (n = 6 mice).

indicating that the increase in dopamine evoked by light is temporal. Interestingly, the contralateral turning behavior in unilateral dopamine-depleted mice induced by dopamine uncaging (Fig 3E) was also gradually reduced after one hour (Fig 3F; contralateral turns at different times, pre: 0 ± 0; 10 min: 1.5 ± 0.5; 30 min: 0.8833 ± 0.3764; 60 min: 0 ± 0) and eventually was switched to ipsilateral turning behavior (Fig 3G; ipsilateral turns at different times, pre: 2.85 ± 0.4506; 10 min: 0 ± 0; 30 min: 0.15 ± 0.1378; 60 min: 1.2 ± 0.4858). These experiments indicate that the local release of dopamine in the striatum of lesioned mice evoke a temporal peak of dopamine that underlies contralateral turning behavior demonstrating that striatal dopamine concentration is tied to the motor effects of dopamine uncaging.

## Tuning of contralateral turning behavior by different light stimulation parameters

It has been suggested that different firing frequencies of dopaminergic neurons could finely tune movements in healthy mice [32–35]. However, it is still unknown if the motor effects of dopamine uncaging in unilateral dopamine-depleted mice are frequency dependent. To characterize the turning behavior in lesioned mice induced by dopamine uncaging at different frequencies we injected RuBi-Dopa into the striatum of the lesioned side of the brain and measured the effects of light uncaging (Fig 4A and S4 Video). We observed that 5 minutes of dopamine release at 1 Hz was unable to induce the contralateral turning behavior previously

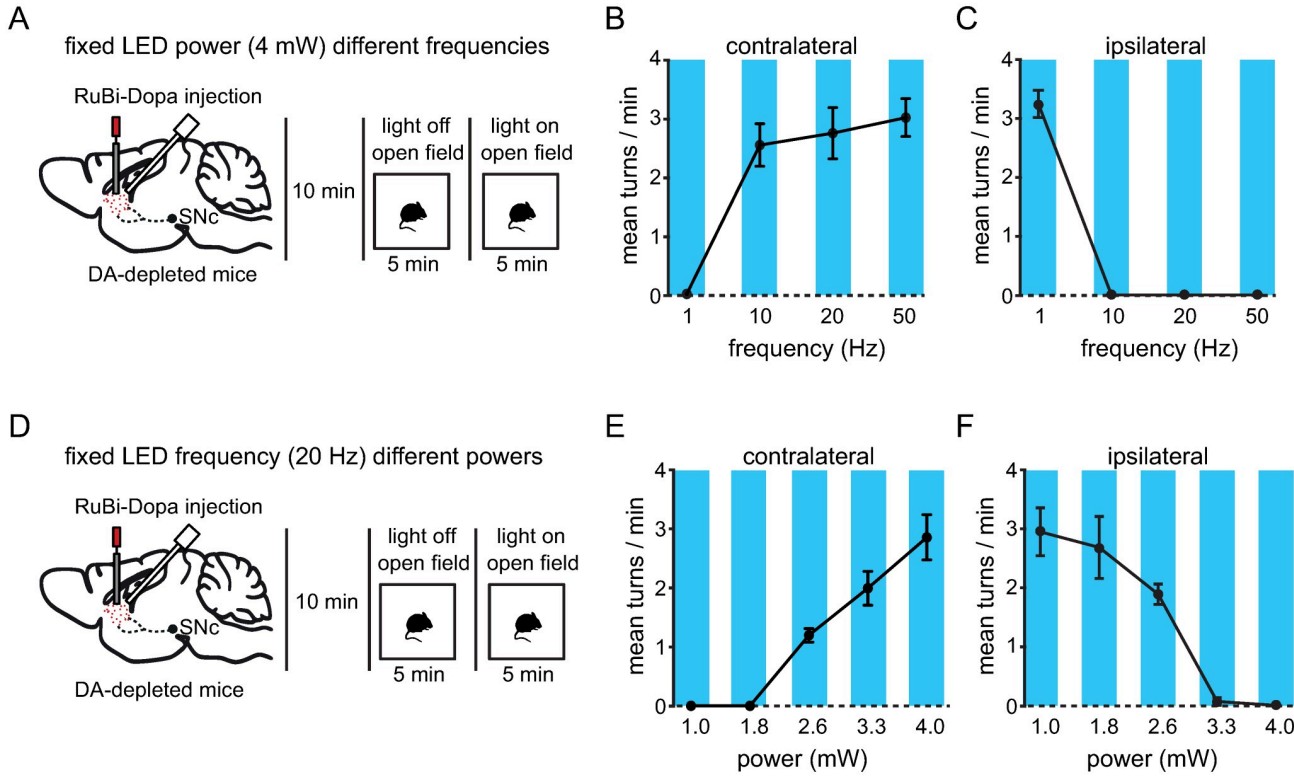

**Fig 4. Light tuning of contralateral turning behavior.** (A) Schematic representation of dopamine uncaging with different light frequencies in unilateral dopamine-depleted mice. (B) Contralateral turning behavior in lesioned mice evoked by different light frequencies keeping the light power constant at 4 mW. Note that contralateral turning behavior is observed with >10 Hz light stimulation. (C) Ipsilateral turning behavior in lesioned mice was suppressed by >10 Hz light stimuli. (D) Schematic representation of dopamine uncaging with different light powers in unilateral dopamine-depleted mice. (E) Contralateral turning behavior in lesioned mice increased as a function of light stimulation power keeping the frequency constant at 20 Hz. (F) Ipsilateral turning behavior in unilateral dopamine-depleted mice was suppressed by >3 mW light power. Time courses display mean ± s.e.m. (n = 3 mice).

observed, whereas 10 Hz, 20 Hz, and 50 Hz light uncaging, keeping the power constant (4 mW), produced contralateral turning behavior in unilateral dopamine-depleted mice (Fig 4B; contralateral turns at 1 Hz: 0 ± 0; 10 Hz: 2.533 ± 0.6252; 20 Hz: 2.733 ± 0.7522; 50 Hz: 3.0 ± 0.5568). Accordingly, the ipsilateral turning behavior was not affected by 1Hz illumination, but was suppressed by 10 Hz, 20 Hz, and 50 Hz light stimuli (Fig 4C; ipsilateral turns at 1 Hz: 3.267 ± 0.4041; 10 Hz: 0 ± 0; 20 Hz: 0 ± 0; 50 Hz: 0 ± 0). These experiments indicate that contralateral turning behavior in unilateral dopamine-depleted mice could be evoked by >10 Hz (4 mW) light uncaging. We next investigated if changes in light power could be used to tune the contralateral turning behavior in dopamine-depleted mice, to do so, we used 20Hz as light frequency and changed the power of the light (Fig 4D). We observed that controlling the output power of the LED and keeping the frequency constant (20 Hz) produced a better modulation of the contralateral turning behavior (Fig 4E; contralateral turns at 1 mW: 0 ± 0; 1.8 mW: 0 ± 0; 2.6 mW: 1.2 ± 0.2; 3.3 mW: 2.0 ± 0.5; 4 mW: 2.867 ± 0.6658) than the change in frequency keeping the power constant since changing the power of illumination was able to control the ipsilateral turning behavior in lesioned mice (Fig 4F; ipsilateral turns at 1 mW: 2.967 ± 0.7095; 1.8 mW: 2.7 ± 0.9165; 2.6 mW: 1.9 ± 0.3; 3.3 mW: 0.1 ± 0.1732; 4 mW: 0 ± 0). These experiments demonstrate that the turning behavior evoked by dopamine uncaging could be tuned by different light stimulation protocols.

## Temporal course of contralateral turning behavior with low and high doses of L-DOPA

The most common therapy for PD consists of the use of L-DOPA as dopamine precursor [4, 13]. It has been shown that the chronic elevation of L-DOPA intake produces motor abnormalities such as L-DOPA induced dyskinesias [14, 36]. So far, our experiments demonstrate that the optical delivery of dopamine, temporarily produced contralateral turning behavior in unilateral dopamine-depleted mice that reflects a temporal increase of dopamine levels. However, it is still unknown how dopamine uncaging relates to a systemic injection of L-DOPA. To investigate the similarities between dopamine uncaging and systemic L-DOPA injection on contralateral turning behavior in lesioned mice we measured the time course of turning behavior at different doses of L-DOPA. A low dose of L-DOPA (Fig 5A) also restricted the movement of mice in an open field arena (Fig 5B) and produced contralateral turning behavior (Fig 5C and S5 Video; contralateral turns at different times, pre: 0.0167 ± 0.0408; 10 min: 2.167 ± 0.6439; 60 min: 1.217 ± 0.4119; 120 min: 0 ± 0) that after 10 minutes resembled the effects of dopamine uncaging (contralateral turns between low dose of L-DOPA at 10 min vs. dopamine uncaging: p = 0.1481; Mann Whitney test; n = 6 mice). However, such L-DOPA induced contralateral turning behavior lasted longer than contralateral turning behavior evoked by dopamine uncaging (Figs 3F & 5C) and was switched to ipsilateral turning behavior after two hours (Fig 5D; ipsilateral turns at different times, pre: 3.967 ± 0.7789; 10 min: 0.15 ± 0.2345; 60 min: 0.1333 ± 0.1506; 120 min: 1.3 ± 0.7823), indicating that dopamine uncaging could be modulated with higher temporal precision than L-DOPA. In contrast, a high dose of L-DOPA (Fig 5E) that has been proved to generated L-DOPA induced dyskinesias in rodents [37, 38] besides producing a restriction in movement (Fig 5F) lasted for more than 2 hours and generated ~400% increment of contralateral turning behavior (Fig 5G and S6 Video; contralateral turns at different times, pre: 0.0167 ± 0.0408; 10 min: 6.083 ± 1.042; 60 min: 8.217 ± 0.8159; 120 min: 2.9 ± 0.6033) that after 10 minutes was significantly higher than dopamine uncaging (contralateral turns between high dose of L-DOPA at 10 min vs. dopamine uncaging: p = 0.0025; Mann Whitney test; n = 6 mice). Furthermore, ipsilateral turning behavior was not recovered after 2 hours when a high dose of L-DOPA was systemically

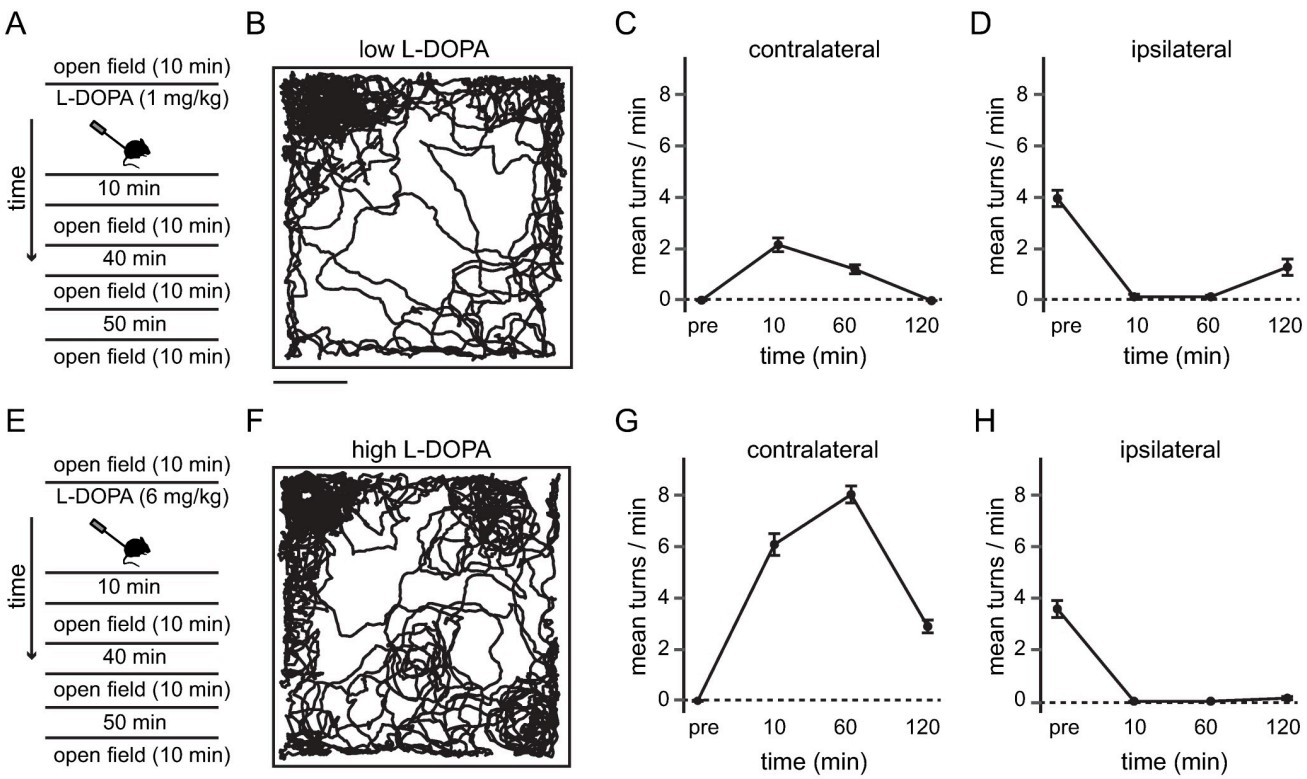

**Fig 5. Contralateral turning behavior with low and high doses of L-DOPA.** (A) Experimental timeline of systemically injected low dose of L-DOPA in unilateral dopamine-depleted mice. (B) Movement trajectory of a representative unilateral dopamine-depleted mouse after 10 minutes of systemic injection of a low dose of L-DOPA (1mg/kg). Note the intermittent restriction of movement due to turning behavior. Scale bar: 10cm. (C) A low dose of L-DOPA induced contralateral turning behavior in lesioned mice for more than 60 minutes. (D) Ipsilateral turning behavior evoked by a low dose of L-DOPA in unilateral dopamine-depleted mice was reinstated after 120 minutes. (E) Experimental timeline of systemically injected high dose of L-DOPA in unilateral dopamine-depleted mice. (F) Movement trajectory of a representative unilateral dopamine-depleted mouse after 10 minutes of systemic injection of a high dose of L-DOPA (6mg/kg). Note the restriction of movement due to turning behavior. Scale bar: 10cm. (G) A high dose of L-DOPA induced contralateral turning behavior in lesioned mice for more than 120 minutes. (H) Ipsilateral turning behavior evoked by a high dose of L-DOPA in unilateral dopamine-depleted mice was suppressed for more than 120 minutes. Time courses display mean ± s.e.m. (n = 6 mice).

injected (Fig 5H; ipsilateral turns at different times, pre: 3.53 ± 0.8035; 10 min: 0 ± 0; 60 min: 0 ± 0; 120 min: 0.1 ± 0.1265). Interestingly, even though mice spend more time in some spots of the open field arena (Fig 5B & 5F) the distance traveled before and after L-DOPA injections was not significantly different (distance traveled before low L-DOPA: 5049 ± 885 cm; distance traveled after low L-DOPA: 4238 ± 708; P = 0.1563; Wilcoxon matched-pairs signed rank test; n = 6 mice; distance traveled before high L-DOPA: 4126 ± 1139 cm; distance traveled after high L-DOPA: 3563 ± 719; P = 0.1563; Wilcoxon matched-pairs signed rank test; n = 6 mice). Control animals (saline injected unilaterally in the SNc) didn't display turning behavior after the systemic injection of low or high L-DOPA. These experiments show that striatal dopamine uncaging in lesioned mice recapitulates the motor effects of a low dose of L-DOPA but for shorter time.

## Effect of dopamine uncaging on synchronized activity of dopamine-depleted striatal populations in anesthetized mice

It has been shown that the unilateral depletion of dopamine generates pathological synchronization of neuronal ensembles in the striatum [7, 39–42]. Such pathological engagement of

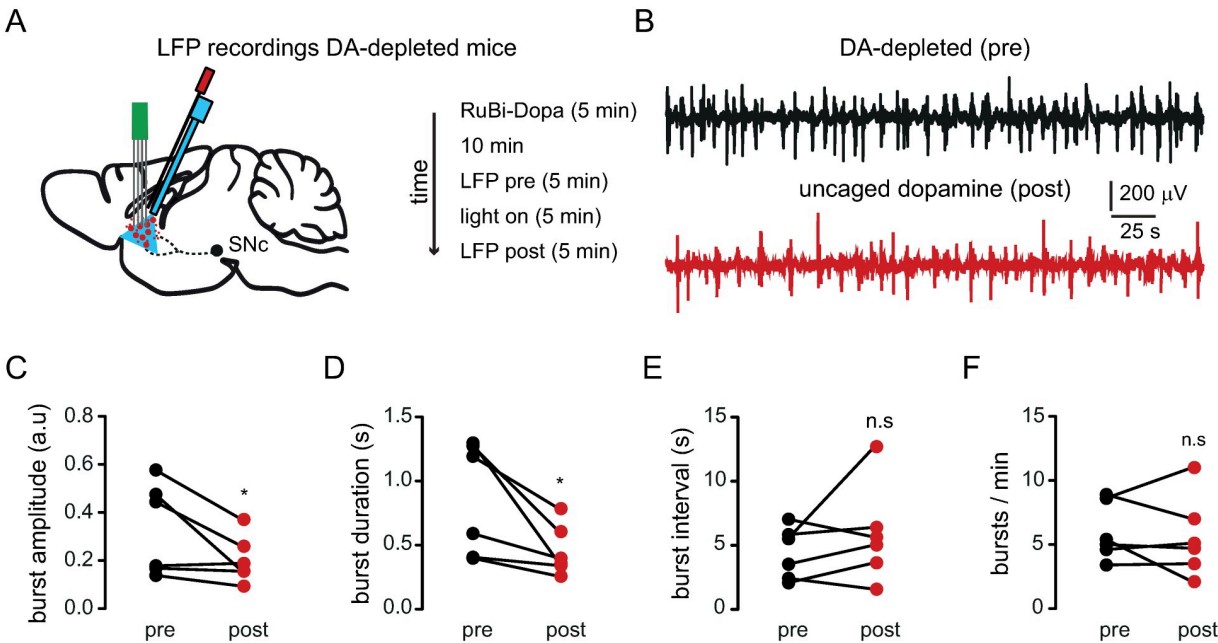

**Fig 6. Effect of dopamine uncaging on pathological population activity.** (A) Schematic representation of striatal dopamine uncaging and LFP recordings in the lesioned side of anesthetized unilateral dopamine-depleted mice. (B) LFPs recorded before (black) and 10 minutes after dopamine uncaging (red). (C) The normalized amplitude of bursting activity in lesioned mice was reduced after dopamine uncaging (*p = 0.0313; n = 6 mice; Wilcoxon matched-pairs signed rank test). (D) The burst duration was decreased by dopamine uncaging (*p = 0.0156; n = 6 mice; Wilcoxon matched-pairs signed rank test). (E) The burst interval showed a tendency to increase after dopamine uncaging that was not significantly different (p = 0.1563; n = 6 mice; Wilcoxon matched-pairs signed rank test). (F) The number of bursts per minute were conserved after dopamine uncaging in unilateral dopamine-depleted mice (p = 0.4219; n = 6 mice; Wilcoxon matched-pairs signed rank test).

striatal activity has also been observed in PD patients [43–45] suggesting that synchronized activity could be used as a biomarker for motor deficits. To investigate the effect of dopamine uncaging on striatal population activity excluding the motor effects, we performed local field potential (LFP) recordings in anesthetized dopamine-depleted mice before and after the optical delivery of dopamine (Fig 6A). We observed that the pathological synchronization of striatal population activity of unilateral dopamine-depleted mice was reduced after dopamine uncaging (Fig 6B), demonstrating a disengagement of striatal neuronal ensembles. The amplitude of bursting activity (see Methods) was reduced significantly after dopamine uncaging (Fig 6C; normalized burst amplitude before uncaging: 0.3305 ± 0.1903 a.u.; normalized burst amplitude after uncaging: 0.2041 ± 0.0980 a.u.), indicating that the transient elevation of dopamine allows the disengagement of pathologically synchronized neurons. Similarly, the duration of bursts was significantly reduced (Fig 6D; burst duration before uncaging: 0.8597 ± 0.4394 seconds; burst duration after uncaging: 0.4577 ± 0.1977 seconds), implying shorter periods of synchronization. On the other hand, the interval between bursts showed a tendency to increase that was not significant (Fig 6E; burst interval before uncaging: 4.409 ± 2.015 seconds; burst interval after uncaging: 5.837 ± 3.771 seconds). Accordingly, the number of bursts per minute before and after dopamine uncaging were not significantly different (Fig 6F; number of bursts per minute before dopamine uncaging: 5.983 ± 2.247; number of bursts per minute after dopamine uncaging: 5.567 ± 3.125). Our results demonstrate that pathological synchronization of population activity observed in dopamine-depleted mice could be disengaged after striatal dopamine uncaging corroborating the rescue of striatal neuronal ensemble dynamics by dopamine [7].

## Discussion

We demonstrated that light-controlled dopamine release in the striatum of unilateral dopamine-depleted mice evokes contralateral turning behavior that was gradually reduced and disappeared after 60 minutes. Importantly, contralateral turning behavior can be tuned by changing the light power and frequency. The motor outcome caused by striatal dopamine uncaging resembles the effect induced by a low concentration of L-DOPA injected systemically but with better temporal control. Furthermore, striatal LFP recordings showed that dopamine uncaging reduced the pathological neuronal synchronization that has been reported in unilateral dopamine-depleted mice.

### Differences between local and global dopamine elevation in PD

In the present study, we focused on the motor effects of dopamine elevation in the striatum of unilateral dopamine-depleted mice. It is known that dopaminergic projections from the mesencephalon have a different distribution throughout the brain that is reflected as different gradients of dopamine across brain nuclei. Thus, it is expected that the dead of dopaminergic neurons observed in PD causes an imbalance of dopaminergic actions that is not homogeneous to all brain nuclei. The systemic elevation of dopamine in treated PD patients could produce unbalanced dopaminergic actions in brain nuclei that are not related to motor control generating non-desired neurological side effects observed in some patients. Furthermore, the fact that the systemic elevation of dopamine has a global effect in the brain makes it difficult to determine the main locus of dopaminergic action to alleviate motor effects in PD. Photopharmacology is particularly suitable to investigate the most effective brain locus of dopaminergic action that could restore motor control avoiding non-desired side effects observed in PD patients. Further experiments are necessary to characterize the effect of dopamine uncaging in other motor and non-motor brain areas using different behavioral paradigms to understand the general role of dopamine in PD.

Compared with optogenetic approaches, dopamine uncaging has more specificity since it has been shown that dopamine is often co-released with different neuromodulators or neurotransmitters [46] making it difficult to dissect the effect of dopamine without other molecules, on the other hand dopamine uncaging doesn't require genetic modifications.

Our results indicate that striatal dopamine uncaging produced a more controlled effect than the systemic injection of apomorphine or the systemic injection of a low dose of L-DOPA that could be explained by their pharmacokinetics and pharmacodynamics. The main limitation, up to now, for the use of caged dopamine in clinical trials is that the light illumination source and delivery system require surgery. The further development of caged compounds that cross the blood-brain barrier and non-invasive methods to uncage them could overcome such difficulties.

### Disengagement of pathological synchronization by dopamine uncaging

According to the classical model of basal ganglia function, it has been proposed that Parkinson's disease is the result of an activity imbalance between the direct and indirect pathways originating in the striatum. However, recent studies demonstrate that the direct and indirect pathways coordinate action selection by concurrently activating or suppressing movements, challenging the notion that next generation therapies for Parkinson's disease should independently modulate the direct or indirect pathways [47–52].

Our electrophysiological population recordings demonstrate an overall reduction in the pathological synchronization observed in anesthetized unilateral dopamine-depleted animals after dopamine uncaging. It has been shown that LFPs not just reflect neuronal ensemble

synchrony but also include information about the afferents around the recording site [53], thus the reduction in synchronization observed could be mediated by the presynaptic and postsynaptic effects of dopamine [54]. Additionally, dopamine loss produces a reduction of the inhibitory feedback connectivity between striatal neurons [55] that disrupts sequential activity patterns between striatal neuronal ensembles [7, 56]. Therefore, dopamine uncaging could also restore recurrent inhibitory connections between striatal neurons giving rise to the reduction of pathological synchronization observed in unilateral dopamine-depleted mice.

It has been shown that the use of selective dopamine receptor agonists induces non-desired side effects [17–19], that could be mediated by dopaminergic receptor activation outside the striatum, suggesting that our approach to control the elevation of dopamine locally in the striatum using light could restore the balance between the direct and indirect striatal pathways by enhancing the activity of the direct pathway through the activation of D1 receptors and reducing the activity of the indirect pathway through the activation of D2 receptors [57].

## Clinical relevance of dopamine uncaging

It has been demonstrated that direct injection of dopamine into the brain has clinical limitations due to the oxidation of dopamine [58] and that intraventricular infusion of anaerobic-dopamine could represent an alternative to alleviate motor deficits in PD [20]. Since caged-dopamine remains stable without light stimulation [25] their limitations due to dopamine oxidation could be minimal making it a good candidate for intracranial pump delivery.

On the other hand, PD patients that have been treated with L-DOPA often develop sudden freezing and motor fluctuations as the disease progresses known as "off" states. Such complications can be alleviated by deep bran stimulation (DBS) or the subcutaneous infusion of apo-morphine [59]. Thus, the future development of delivery systems for dopamine uncaging into the brain could be used to reduce such episodes.

Compared to DBS that requires the use of electrodes, optical fibers are prone to less degradation than electrodes into the brain [60], suggesting that photopharmacology [22] could represent a long lasting solution compared to electrical stimulation.

Our experiments demonstrate that one striatal injection of RuBi-Dopa irradiated for 5 minutes could evoked contralateral turning behavior that gradually disappears after 60 minutes. The fact that contralateral turning behavior reflects striatal dopamine levels (Fig 3), and that contralateral turning behavior could be modulated by tuning the light (Fig 4), suggest that a fine control of dopamine fluctuations can be achieved modulating the duration, power, and frequency of the light. To mimic physiological dopamine fluctuations, fluorescent dopamine indicators [61] could be used to characterize naturalistic dopamine changes with high spatio-temporal resolution during behavioral tasks or brain states and then closed loop systems could be designed to control dopamine uncaging by light tuning. Naturalistic release of dopamine using a closed-loop delivery system could have a long-term rescue of the devastating motor deficits observed in Parkinson's disease without the non-desired effects of sudden and chronic dopamine elevations caused by traditional pharmacology [62, 63].

Before using this approach in clinical trials, it would be necessary to study the absorption, distribution, metabolism, and toxicity of ruthenium-based caged compounds in the brain to avoid possible inconveniences of dopamine uncaging.

Finally, our results suggest that local release of dopamine could improve motor abnormalities and avoid non-desired side effects observed when dopaminergic agonists are injected systemically since dopamine uncaging could be targeted to motor brain areas.

## Supporting information

**S1 Video. Unilaterally dopamine depleted mouse in an open field.** Note ipsilateral turning behavior. Video shows 10 minutes of recording (speed up 10X).
(MP4)

**S2 Video. Unilaterally dopamine depleted mouse in an open field 10 minutes after systemic apomorphine injection.** Note contralateral turning behavior. Video shows 10 minutes of recording (speed up 10X).
(MP4)

**S3 Video. Unilaterally dopamine depleted mouse in an open field 10 minutes after striatal dopamine uncaging.** Note contralateral turning behavior. Video shows 10 minutes of recording (speed up 10X).
(MP4)

**S4 Video. Unilaterally dopamine depleted mouse in an open field during striatal dopamine uncaging.** Note contralateral turning behavior caused by photo-stimulation at 20 Hz (4 mW). Video shows 50 seconds of recording before photo-stimulation and 5 minutes of dopamine uncaging during blue light illumination (speed up 10X).
(MP4)

**S5 Video. Unilaterally dopamine depleted mouse in an open field 10 minutes after systemic L-DOPA injection at low concentration.** Note contralateral turning behavior. Video shows 10 minutes of recording (speed up 10X).
(MP4)

**S6 Video. Unilaterally dopamine depleted mouse in an open field 10 minutes after systemic L-DOPA injection at high concentration.** Note contralateral turning behavior. Video shows 10 minutes of recording (speed up 10X).
(MP4)

**S7 Video. Control mouse in an open field 10 minutes after striatal dopamine uncaging.** Note the lack of ipsilateral or contralateral turning behavior. Video shows 10 minutes of recording (speed up 10X).
(MP4)

**S1 Fig. Lack of contralateral turning behavior in control mice.** (A) Experimental timeline of unilateral striatal photo-stimulation in control mice. (B) Movement trajectory of a representative control mouse placed on an open field arena after striatal injection of saline followed by photo-stimulation. Scale bar: 10cm. (C) Striatal photo-stimulation in control mice doesn't evoke contralateral turning behavior. (D) Experimental timeline of unilateral striatal injection of RuBi-Dopa in control mice. (E) Movement trajectory of a representative control mouse placed on an open field arena after striatal dopamine uncaging. Scale bar: 10cm. (F) Dopamine uncaging in control mice doesn't evoke contralateral turning behavior. (G) Experimental timeline of systemically injected apomorphine in control mice. (H) Movement trajectory of a representative control mouse placed on an open field arena after systemic injection of apomorphine. Scale bar: 10cm. Note that the mouse moves close to the border of the open field arena. (I) Systemic injection of apomorphine in control mice doesn't evoke contralateral turning behavior.
(TIF)

**S1 Dataset.**
(XLSX)

## Acknowledgments

We thank Vladimir Calderon, Elsa Nydia Hernández Rios, Ericka A. de los Rios Arellano, Martín Garcia Servin, Alejandra Castilla Leon and Deysi Gasca Martinez for technical assistance. We thank Rafa Yuste for the generous donation of materials and reagents.

## Author Contributions

**Conceptualization:** Miguel A. Zamora-Ursulo, Luis A. Tellez, Nadia Saderi, Luis Carrillo-Reid.

**Data curation:** Miguel A. Zamora-Ursulo, Job Perez-Becerra.

**Formal analysis:** Miguel A. Zamora-Ursulo, Job Perez-Becerra.

**Funding acquisition:** Luis Carrillo-Reid.

**Investigation:** Miguel A. Zamora-Ursulo, Job Perez-Becerra, Luis A. Tellez.

**Methodology:** Miguel A. Zamora-Ursulo, Job Perez-Becerra, Luis A. Tellez.

**Resources:** Luis A. Tellez, Nadia Saderi.

**Software:** Miguel A. Zamora-Ursulo.

**Supervision:** Nadia Saderi, Luis Carrillo-Reid.

**Visualization:** Job Perez-Becerra, Luis Carrillo-Reid.

**Writing – original draft:** Luis Carrillo-Reid.

**Writing – review & editing:** Luis Carrillo-Reid.

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
