## [Decision Letter · Decision Letter 0]

19 Jul 2023

PONE-D-23-15033Reversal of pathological motor behavior in a model of Parkinson’s disease by dopamine uncagingPLOS ONE

Dear Dr. Carrillo-Reid,

Thank you for submitting your manuscript to PLOS ONE. After careful consideration, we feel that it has merit but does not fully meet PLOS ONE’s publication criteria as it currently stands. Therefore, we invite you to submit a revised version of the manuscript that addresses the points raised during the review process.

We look forward to receiving your revised manuscript.

Kind regards,

Luca Aquili

Academic Editor

PLOS ONE

“We thank Vladimir Calderon, Elsa Nydia Hernández Rios, Ericka A. de los

Rios Arellano, Martín Garcia Servin, Alejandra Castilla Leon and Deysi Gasca Martinez for

technical assistance. We thank Rafa Yuste for the generous donation of materials and reagents.

This research was supported by grants from CONACYT (CF6653, CF154039) and UNAM[1]465 DGAPA-PAPIIT (IA201421, IA201819, IN213923) to L.C-R. MA.Z-U. participated in this work in

partial fulfillment of the requirements for the Ph.D. degree in Basic Biomedical Sciences at the

Universidad Autonoma de San Luis Potosi graduate scholarship from CONACYT (770504).”

“This research was supported by grants from CONACYT (CF6653, CF154039) and UNAM-DGAPA-PAPIIT (IA201421, IA201819, IN213923) to L.C-R.

MA.Z-U. participated in this work in partial fulfillment of the requirements for the Ph.D. degree in Basic Biomedical Sciences at the Universidad Autonoma de San Luis Potosi graduate scholarship from CONACYT (770504).

The funders had no role in study desing, data collection and analysis, decision to publish, or preparation of the manuscript.”

Reviewers' comments:

Reviewer's Responses to Questions

**Comments to the Author**

1. Is the manuscript technically sound, and do the data support the conclusions?

Reviewer #1: Yes

Reviewer #2: Partly

2. Has the statistical analysis been performed appropriately and rigorously? 

Reviewer #1: Yes

Reviewer #2: Yes

3. Have the authors made all data underlying the findings in their manuscript fully available?

Reviewer #1: Yes

Reviewer #2: Yes

4. Is the manuscript presented in an intelligible fashion and written in standard English?

Reviewer #1: Yes

Reviewer #2: Yes

5. Review Comments to the Author

Reviewer #1: The paper entitled “Reversal of pathological motor behavior in a model of Parkinson’s disease by dopamine uncaging” by Zamora-Ursulo et al. is interesting. It describes the behavioral, biochemical, and electrophysiological effects of RuBi-Dopa in DA-depleted mice at three weeks post-6-OHDA unilateral injection in the substantia nigra.

Given the complexity of the actions of DA in the brain, with particular attention to Parkinson’s disease, this referee is particularly impressed by the fact that light-uncaged DA in the striatum has motor effects that could mimic those exerted by the system injection of DA agonists.

There is also the belief that systemic DA agonists (direct or indirect) could act on extrastriatal sites (e.g., Substantia nigra) to alleviate motor impairment.

Therefore we could expect the less intense contralateral turning to uncaged DA even at a high degree of light stimulation, compared to that caused by systemic apomorphine or levodopa. The authors might consider this point.

There is, of course, also an explanation for why the effects of systemic levodopa last longer the those caused by local uncaged DA. The rate of its arrival in the brain, its transformation in DA in the remaining DAergic terminals, and degradation account for the different time courses.

In the possible future use of this technique, I would suggest it be effective in combatting the sudden freezing of PD patients

Minor

If the cannulae were implanted almost three weeks before the infusion experiments, how did the authors not let them be plugged?

Did the authors have data regarding the unilateral local application of RuBi-Dopa and the uncaged DA in the normal striatum?

Reviewer #2: In the present study, the authors reported that stereotaxic administration of caged molecule RuBi-Dopa and its irradiation by blue light resulted in controlled increase in the striatal dopamine level and reduced the contralateral turning behavior in 6-OHDA induced PD mice.

The present piece of work appears interesting and might make significant contribution in the development of smart and cutting-edge treatment strategies for PD. However, some issues have been noticed which require modifications before further consideration and acceptance.

Comments-

1. The abstract appears to be unnecessarily long and does not contain concise and wholesome information about the entire work. The abstract must be modified in such a way that readers do not feel off-track while reading it. The abstract must contain a wholesome gist of introduction, methodology, results, and final conclusion and future perspective.

2. In the Introduction section, the authors must write more about caged molecules, their physical and chemical aspects, working mechanism, and their contribution so far in the field of medicine and PD.

3. The figures provided require proper alignment.

4. The authors must provide more information about what plausible side effects or inconvenience to the recipients this presented approach of PD treatment can bring.

5. The novelty of the present work must be highlighted in a more justifiable way.

6. PLOS authors have the option to publish the peer review history of their article (what does this mean?). If published, this will include your full peer review and any attached files.

Reviewer #1: **Yes: **Nicola Biagio Mercuri

Reviewer #2: **Yes: **Anupom Borah

---

## [Author Response · Author response to Decision Letter 0]

4 Aug 2023

We thank the Reviewers and Editor for their helpful comments and suggestions that we considered have strengthened the impact of our study. We followed all the recommendations changing the text accordingly and adding supporting information to show the experiments. We also added 2 references to support the new text included in the new version of the manuscript (Refs 23 and 59). All the changes to the manuscript are highlighted in red in the Track version. 

For clarity, our responses to reviewers are highlighted in bold style. 

Point by point responses to Reviewer #1

Reviewer #1: The paper entitled “Reversal of pathological motor behavior in a model of Parkinson’s disease by dopamine uncaging” by Zamora-Ursulo et al. is interesting. It describes the behavioral, biochemical, and electrophysiological effects of RuBi-Dopa in DA-depleted mice at three weeks post-6-OHDA unilateral injection in the substantia nigra.

Given the complexity of the actions of DA in the brain, with particular attention to Parkinson’s disease, this referee is particularly impressed by the fact that light-uncaged DA in the striatum has motor effects that could mimic those exerted by the system injection of DA agonists. There is also the belief that systemic DA agonists (direct or indirect) could act on extrastriatal sites (e.g., Substantia nigra) to alleviate motor impairment.

Response: We appreciate that Reviewer #1 finds our manuscript interesting and our results of striatal dopamine uncaging impressing. We are excited to use this approach in the future to understand the role of dopamine in different brain areas. We added some lines in the discussion (lines 497-513) considering this comment. 

Therefore we could expect the less intense contralateral turning to uncaged DA even at a high degree of light stimulation, compared to that caused by systemic apomorphine or levodopa. The authors might consider this point.

Response: We measured the turning behavior in most of the experiments after 10 minutes of systemic injection of dopamine or dopamine uncaging. In such temporal window we observed less intense contralateral turning behavior evoked by DA uncaging. Interestingly, the contralateral turning behavior was more intense when we measured it at the time of dopamine uncaging at high stimulation parameters (Fig 4 and S4 Video). 

There is, of course, also an explanation for why the effects of systemic levodopa last longer the those caused by local uncaged DA. The rate of its arrival in the brain, its transformation in DA in the remaining DAergic terminals, and degradation account for the different time courses.

Response: We agree with the reviewer and add some lines in the discussion (lines 519-520) of the new version of the manuscript considering this comment.

In the possible future use of this technique, I would suggest it be effective in combatting the sudden freezing of PD patients.

Response: We really appreciate this suggestion. We added a new paragraph in the discussion (lines 554-558) to highlight this possible future application.

Minor

If the cannulae were implanted almost three weeks before the infusion experiments, how did the authors not let them be plugged?

Response: The cannulae used in the experiments have a removable dummy protective cap to avoid clogging. We added lines 123-124 in the Methods section to make this point clear.

Did the authors have data regarding the unilateral local application of RuBi-Dopa and the uncaged DA in the normal striatum?

Response: Yes, we have such data and we included new supporting information to show that striatal dopamine uncaging doesn’t evoke contralateral turning behavior (S1 Fig and S7 Video) in control mice. 

Point by point responses to Reviewer #2

Reviewer #2: In the present study, the authors reported that stereotaxic administration of caged molecule RuBi-Dopa and its irradiation by blue light resulted in controlled increase in the striatal dopamine level and reduced the contralateral turning behavior in 6-OHDA induced PD mice.

The present piece of work appears interesting and might make significant contribution in the development of smart and cutting-edge treatment strategies for PD. However, some issues have been noticed which require modifications before further consideration and acceptance.

Response: We appreciate that Reviewer #2 considers our manuscript interesting with potential contribution to the treatment of PD. 

Comments-

1. The abstract appears to be unnecessarily long and does not contain concise and wholesome information about the entire work. The abstract must be modified in such a way that readers do not feel off-track while reading it. The abstract must contain a wholesome gist of introduction, methodology, results, and final conclusion and future perspective.

Response: We modified the abstract following the Reviewer recommendations.

2. In the Introduction section, the authors must write more about caged molecules, their physical and chemical aspects, working mechanism, and their contribution so far in the field of medicine and PD.

Response: We added new lines in the introduction describing the general characteristics of caged compounds (lines 63-77; 81-82). So far, the focus of the potential clinical use of caged compounds have been restricted to cancer or retinal applications. However, there’s not a current application in the field of medicine. 

3. The figures provided require proper alignment.

Response: We modified the alignment of the figures in the new version of the manuscript using the online software recommended by the journal (PACE).

4. The authors must provide more information about what plausible side effects or inconvenience to the recipients this presented approach of PD treatment can bring.

Response: We added a new paragraph in the discussion (lines 574-576) highlighting the necessary considerations and possible side effects of implementing this approach to clinical trials.

5. The novelty of the present work must be highlighted in a more justifiable way.

Response: We thank the Reviewer for this comment. We have modified the Abstract, Introduction and Discussion (Red text in Track version) to highlight our results according to the experiments shown in the present work.

---

## [Editor Report · Decision Letter 1]

7 Aug 2023

Reversal of pathological motor behavior in a model of Parkinson’s disease by striatal dopamine uncaging

PONE-D-23-15033R1

Dear Dr. Carrillo-Reid,

We’re pleased to inform you that your manuscript has been judged scientifically suitable for publication and will be formally accepted for publication once it meets all outstanding technical requirements.

Kind regards,

Luca Aquili

Academic Editor

PLOS ONE
---

## [Editor Report · Acceptance letter]

10 Aug 2023

PONE-D-23-15033R1 

Reversal of pathological motor behavior in a model of Parkinson’s disease by striatal dopamine uncaging 

Dear Dr. Carrillo-Reid:

I'm pleased to inform you that your manuscript has been deemed suitable for publication in PLOS ONE. Congratulations! Your manuscript is now with our production department. 

Kind regards, 

on behalf of

Dr. Luca Aquili 

Academic Editor

PLOS ONE